

# The effect of climate change on the distribution of a tropical zoanthid (*Palythoa caribaeorum*) and its ecological implications

Leonardo M. Durante[1,2,*], Igor C.S. Cruz[1,3] and Tito M.C. Lotufo[1,*]

[1] Oceanographic Institute, University of Sao Paulo, São Paulo, SP, Brazil
[2] University of Otago, Dunedin, New Zealand
[3] Universidade Federal da Bahia, Salvador, BA, Brazil
[*] These authors contributed equally to this work.

Corresponding author
Tito M.C. Lotufo, tmlotufo@usp.br

## ABSTRACT

*Palythoa caribaeorum* is a zoanthid often dominant in shallow rocky environments along the west coast of the Atlantic Ocean, from the tropics to the subtropics. This species has high environmental tolerance and is a good space competitor in reef environments. Considering current and future scenarios in the global climate regime, this study aimed to model and analyze the distribution of *P. caribaeorum*, generating maps of potential distribution for the present and the year 2100. The distribution was modeled using maximum entropy (Maxent) based on 327 occurrence sites retrieved from the literature. Calcite concentration, maximum chlorophyll-*a* concentration, salinity, pH, and temperature range yielded a model with the smallest Akaike information criterion (2649.8), and were used in the present and future distribution model. Data from the HadGEM2-ES climate model were used to generate the projections for the year 2100. The present distribution of *P. caribaeorum* shows that parts of the Brazilian coast, Caribbean Sea, and Florida are suitable regions for the species, as they are characterized by high salinity and pH and small temperature variation. An expansion of the species' distribution was forecast northward under mild climate scenarios, while a decrease of suitable areas was forecast in the south. In the climate scenario with the most intense changes, *P. caribaeorum* would lose one-half of its suitable habitats, including the northernmost and southernmost areas of its distribution. The Caribbean Sea and northeastern Brazil, as well as other places under the influence of coastal upwellings, may serve as potential havens for this species.

# INTRODUCTION

*Palythoa caribaeorum* (Duchassaing & Michelotti, 1860) is a sessile colonial anthozoan common along the west coast of the Atlantic Ocean (*Kemp et al., 2006*). This species can be found from Florida in the US (*Kemp et al., 2006*) to Arvoredo Island in Brazil (*Bouzon, Brandini & Rocha, 2012*), as well as around many oceanic islands in the Atlantic, such

as Bermuda (*Lesser et al., 1990*) and the São Pedro e São Paulo archipelago (*Edwards & Lubbock, 1983*). Its distribution is not limited to the western side of the Atlantic Ocean, however, with records in the Cape Verde islands (*Reimer, Hirose & Wirtz, 2010*). This broad geographic distribution is possible because *P. caribaeorum* has a great tolerance to a wide range of environmental conditions (*Sebens, 1982*). Although studies on *P. caribaeorum* larvae are lacking, it is expected that its biology would be similar to its sister species, *Palythoa turbeculosa* (Esper 1791), which has symbiotic larvae with planktonic habits and a larval stage up to 170 days long (*Polak et al., 2011*; *Ryland, 1997*).

It is found on hard bottom environments, from the intertidal zone to depths up to 12 m (*Sebens, 1982*), where it can be dominant (*Monteiro et al., 2008*). Colonies have a mat format usually occupying large areas along the substratum (*Sebens, 1982*; *Acosta, 2001*; *Silva et al., 2015*). The animal's color varies from yellow and brown to bronze, easily identifiable in the field (Fig. 1A).

*P. caribaeorum* is a mixotrophic organism, hosting symbiotic dinoflagellates (*Symbiodinium*) inside its tissues (*Sebens, 1977*; *Suchanek & Green, 1981*) and feeding on planktonic organisms, which contributes to the energy flux from plankton to benthic environments (*Sorokin, 1991*; *Santana et al., 2015*). Few studies have highlighted the importance of *P. caribaeorum* in association with other invertebrates (*Pérez, Vila-Nova & Santos, 2005*), reef dynamics (*Silva et al., 2015*), or the enrichment of the bottom fish community (*Mendonça-Neto et al., 2008*). However, a study conducted in the Todos os Santos Bay in Brazil (*Cruz et al., 2015b*) showed that the increased abundance of *Palythoa* sp. decreased habitat heterogeneity, and consequently the richness of fish and benthic organisms, changing the local trophic structure in that region.

*P. caribaeorum* is capable of overgrowing and thus adversely affecting the growth and recruitment of several organisms (*Suchanek & Green, 1981*; *Castro et al., 2012*) such as hard corals, hydrocorals, and other zoanthids, showing great efficiency competing for space. *P. caribaeorum* also has a fast growth rate estimated to vary between 0.04-0.15 mm per day (*Bastidas & Bone, 1996*; *Silva et al., 2015*). In general, zoanthids can dominate in areas where the environmental factors make it difficult for the settlement of scleractinian corals (*Fautin, 1988*; *Cruz et al., 2015a*), often becoming important as the main benthic component (*Mendonça-Neto et al., 2008*) and nursery habitat for other invertebrates (*Pérez, Vila-Nova & Santos, 2005*). *Suchanek & Green (1981)* and *Sebens (1982)* claimed that predation does not seem to be an ecological element controlling the distribution and abundance of *P. caribaeorum,* probably due to defense strategies related to nematocysts and palytoxin that are found in the animal's tissue (*Hines & Pawlik, 2011*). However, *Stampar, Silva & Luiz Jr (2007)* reported a case of a Hawksbill turtle, *Eretmochelys imbricata*, preying colonies of *P. caribaeorum* in southeastern Brazil.

Regarding its ecological role, investigating the current and future distribution of *P. caribaeorum* will yield a prognosis of areas likely to experience changes in their benthic communities because of the presence or absence of *P. caribaeorum*. Assessing how species and biological systems are affected by climate change as well as the generation of forecast models are increasingly needed by decision makers and environmental managers (*Gutt et al., 2012*).
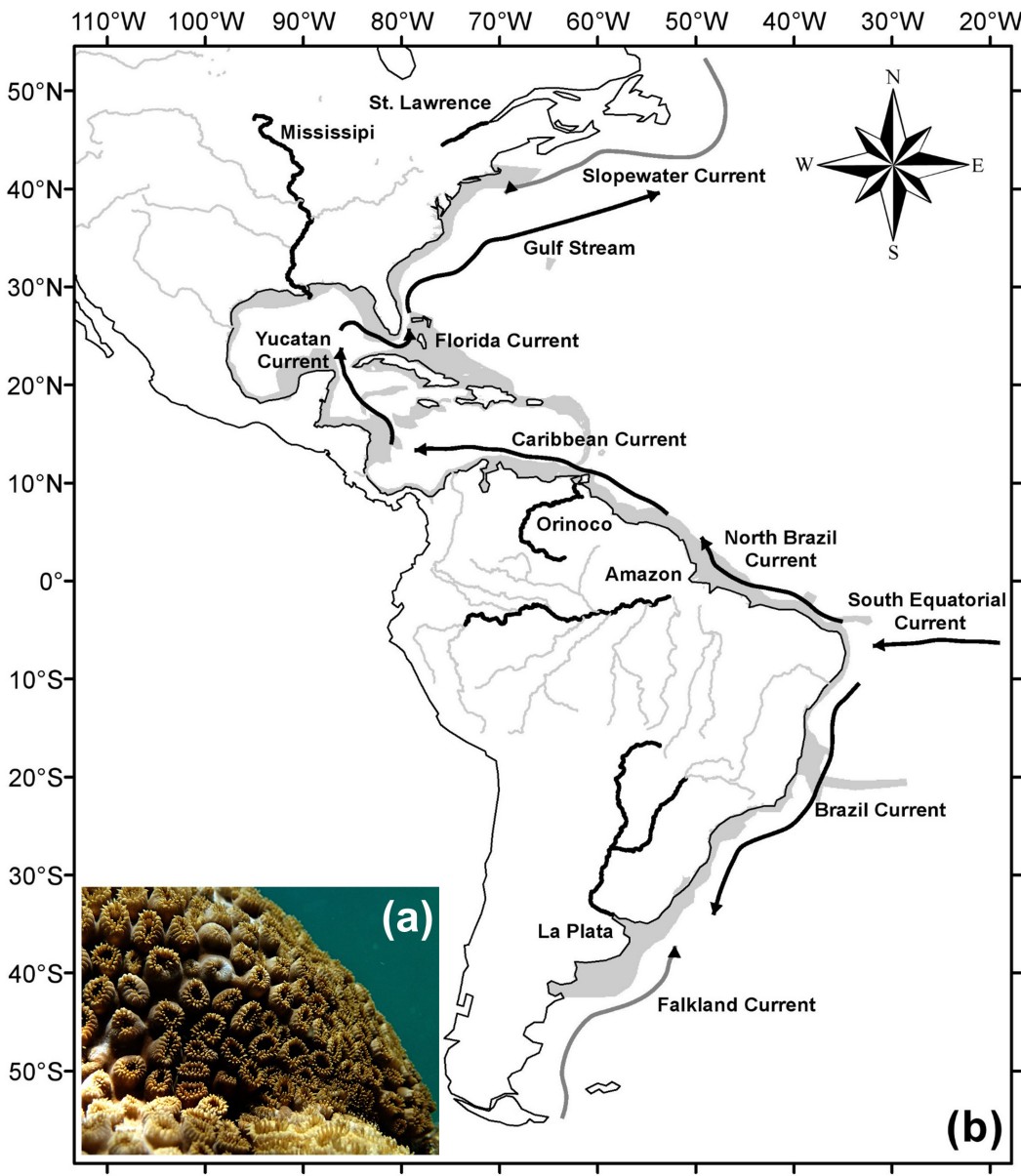

**Figure 1** **Study area and major currents and rivers.** Underwater picture of (A) a *Palythoa caribaeo-rum* colony and (B) the study area (gray) with main rivers (black), adapted from the map "World Major Rivers" from ESRI; the schema of the main cold surface currents (gray) and warm surface currents (black) is adapted from *Talley et al. (2011)*. Photo credit: Leonardo Durante.

Species distribution models (SDM) are largely used in ecology, biogeography, and conservation biology to help solve practical problems in these fields (*Guisan & Thuiller, 2005*; *Elith et al., 2011*; *Acosta et al., 2016*). One of the main assumptions of SDM is that the environment is in equilibrium with its species, whose distribution is determined by environmental variability (*Araújo & Pearson, 2005*), excluding biological interactions,

dispersion barriers, and fast or small-scale environmental disturbances (*De Marco, Diniz-Filho & Bini, 2008*). Temporal and climatic variations also interfere in the modeling, and environmental data must be gathered from the same period in which the species was recorded, making it difficult to acquire large amounts of data for mobile organisms. In this scenario, *P. caribaeorum* is a good candidate for the use of SDM.

The accelerating rates of environmental change pose a challenge to most organisms living in shallow waters throughout the world's oceans. At a global scale, between 1971 and 2010, the warming rate of the oceans' surface waters reached 0.13 °C per decade (*IPCC, 2014*). Since 1950, coastal regions with high evaporation rates are becoming saltier; and for the same period, the pH of surface waters has decreased by 0.1 (*IPCC, 2014*). The forecast for the year 2050 shows a redistribution of marine organisms as a function of the increase in temperature. This redistribution is expected to be toward mid-latitude zones, increasing species richness in these areas while decreasing richness in the tropics due to extinction, migration, and reduction in genetic diversity (*Fields et al., 1993*). However, it is still unknown for sure how the effects of climate change, at the population level, will affect the environment at large ecological scales (*Harley et al., 2006*; *Thomas et al., 2017*).

This study aimed to investigate and model the distribution of *P. caribaeorum* along the western coast of the Atlantic Ocean, generating potential distribution maps for the present and the year 2100 based on forecast climate models. The maps were used to characterize the species' habitat and identify areas where suitability was retained, lost, or gained for *P. caribaeorum* over time and under different climate scenarios.

## MATERIALS AND METHODS

### Study area

The study area comprised the western Atlantic Ocean (from −42.5° to 42.5° latitude), between 2 m of altitude and 100 m of depth (Fig. 1B), hence extending beyond the current known distribution of *P. caribaeorum*. The oceanographic features of this region are influenced by the South Equatorial Current, from the east, which is divided into two components: northward (North Brazil Current) and southward along the coast (Brazil Current). The North Brazil Current flows toward the northwest, passing by the Caribbean Sea, Yucatan Channel, and Gulf of Mexico before merging with the Antilles Current in the Florida Strait to form the Gulf Stream, which flows toward Europe. Due to the strong tropical component, the region between 20° S and 30° N is strongly influenced by waters heated in equatorial regions (*Ekman, 1953*). In contrast, regions closer to the boundaries of this study are influenced by both tropical and colder currents, such as the Slopewater Current from the Labrador Current in the north (*Churchill & Berger, 1998*) and the Malvinas Current in the south (*Talley et al., 2011*). The study region is also influenced by large river discharges, mainly from the La Plata, Amazon, Orinoco, Mississippi, and Saint Lawrence rivers (Fig. 1B).

### Biological data mining and filtering

An extensive literature review was performed to obtain the largest possible number of studies with occurrence data for *P. caribaeorum* (Supplemental Information 1). Occurrence
**Table 1** Details about the variables retrieve from Bio-Oracle dataset and the layers kept in the final distribution model for *P. caribaeorum.*

| Variables | Unit | Measurement | Measurement transformations | Number of layers | Kept in the final model |
|---|---|---|---|---|---|
| Calcite concentration | mol/m$^3$ | Remote sensor | Average | 1 | Average concentration |
| Chlorophyll-*a* concentration | mg/m$^3$ | Remote sensor | Average, minimum, maximum and range | 4 | Maximum concentration |
| Cloud coverage | % | Remote sensor | Average, minimum and maximum | 3 | – |
| Difusive atenuation coeficient (490 nm) | 1/m | Remote sensor | Average, minimum and maximum | 3 | – |
| Photosynthetic available radiation | Einstein/m$^2$/day | Remote sensor | Average and maximum | 2 | – |
| Air temperature above sea surface | °C | Remote sensor | Average, minimum, maximum and range | 4 | – |
| Sea surface temperature | °C | Remote sensor | Average, minimum, maximum and range | 4 | Range |
| Dissolved oxygen | ml/l | In situ | Diva interpolation | 1 | – |
| Nitrate | μmol/l | In situ | Diva interpolation | 1 | – |
| pH | – | In situ | Diva interpolation | 1 | Diva interpolation |
| Phosphate | μmol/l | In situ | Diva interpolation | 1 | – |
| Salinity | PSU | In situ | Diva interpolation | 1 | Diva interpolation |
| Silicate | μmol/l | In situ | Diva interpolation | 1 | – |

data recorded by the authors of this study and from global databases, such as WORMS, OBIS, and GBIF, were also used, totaling 327 occurrence points for the species. The three identified synonyms (*Reimer, 2015*)—*Palythoa caribaea* Duerden, 1898, *Palythoa caribbaea* Goreau, 1959, and *Palythoa caribdea* Carballeira et al., 1998—were included.

Occurrence points found on land or at depths deeper than 100 m were automatically excluded, as were those localized in freshwater environments. Because sampled areas and methodologies varied among study sites, only one occurrence per pixel (5 arcminutes), based on the environmental variables dataset, was kept for analysis (*Elith & Leathwick, 2009*). This point was placed in the middle of each pixel, disregarding the exact occurrence point. After that, to avoid bias originating from densely sampled sites, the geographic filter "OccurrenceThinner" version 1.04 (*Verbruggen, 2012*), with thresholds of 0.5 and 1, was used, resulting in the final dataset.

## Selecting environmental variables, the regulation multiplier value, and the running Maxent

Environmental and bathymetric data, with resolutions of 5 and 1 arcminute, were obtained from the Bio-Oracle (*Tyberghein et al., 2012*) dataset and the ETOPO1 Global Relief Model, respectively (*Amante & Eakins, 2009*). Twenty-seven layers of environmental data and their transformations available on the Bio-Oracle dataset were used in the analyses (Table 1). Each layer was cropped using ArcGIS 10.2 to only include areas within 2 m altitude and 100 m depth.
The habitat modeling was done using the Maxent 3.3.3 program (*Phillips, Anderson & Schapire, 2006*). Maxent has the advantage of using only occurrence points, instead of occurrence and absence/pseudo-absence, considering the lack of this type of data, especially for marine organisms. Maxent also demonstrates good performance identifying suitable habitats (*Phillips, Anderson & Schapire, 2006*; *Reiss et al., 2011*; *Meißner et al., 2014*) and predicting novel distributions for species (*Costa et al., 2012*; *Acosta et al., 2016*) when compared with other software.

All the features available in Maxent were used during the modeling (standard set up). To choose the environmental variables and the regulation multiplier (*Rm*) value, the "MaxentVariableSelection" package (*Jueterbock, 2016*) for R 3.3.1 (*R Core Team, 2015*) was used, with a contribution threshold of 4% and Pearson's correlation coefficient of 0.9. A series of runs were conducted, varying the *Rm* from 0.5 to 10 with increments of 0.5, and were then assessed using the Akaike Information Criterion (AIC). The environmental variables and *Rm* of the model with a lower AIC were used to run the final Maxent model. The AIC tends to choose models that sort the most important variables, as well as models that better estimate environmental suitability (*Warren et al., 2014*), and therefore behave better when projected onto novel climate scenarios than models chosen using the Area Under the Receiver Operator Curve (AUC) value (*Warren & Seifert, 2011*). The AIC selects the optimal and most parsimonious model, taking into account its complexity (*Bozdogan, 1987*). Thirty percent of the data were randomly sampled to test the model along each of the 100 replicates, resulting in an average for all replicates. In all the analyses, 10,000 random points were used as the background along the whole study area.

## Climate model data

Data from the HadGEM2-ES model (*Collins et al., 2011*) retrieved from the Coupled Model Intercomparison Project (CMIP5) were used on the projections for the year 2100 under three different Representative Concentration Pathway (RCP) climate scenarios: low, middle, and high radiative forcing of 2.6, 4.5, and 8.5 W/m$^2$, respectively (*Van Vuuren et al., 2011*). The RCPs are related to retention of energy by the Earth's surface due to accumulation of greenhouse gases from human activities, therefore higher numbers represent a greater anthropogenic impact on our biosphere. The HadGEM2-ES climate model was freely available and provided outputs of the main interest variables and high resolution in tropical areas under the RCP climate scenarios. When there were no available data for the projections, the layers were kept unchanged. NetCDF files from the climate model were interpolated using the bilinear interpolation "interp2" of Matlab 7.14, and were then cropped and converted to ASC format.

## Projecting the results onto maps

Maps were generated using the cumulative likelihood of species occurrence obtained from Maxent, which can be interpreted as the sum of the relative occurrence rate of the species in one pixel and other pixels with the same or lower values of the relative occurrence rate (ROR). This type of output is best for drawing species distribution boundaries for different climate scenarios (*Merow, Smith & Silander, 2013*) and also has a better visual presentation.

For the binary maps, the threshold equal to the maximum sensitivity plus specificity (max SSS) was chosen to detect suitable habitats for *P. caribaeorum*. This threshold improves the distinction between occurrence and absence sites, as well as between occurrence and background, also selecting the same threshold for both occurrence-only and occurrence-absence models (*Liu, Newell & White, 2015*).

Because *P. caribaeorum* is distributed along coastal zones, the final results were cropped to depths up to 30 m, better defining its real habitat. The map of the Marine Ecoregions of the World (*Spalding et al., 2007*) was used as a reference for the results and discussion of specific areas (Fig. 2).

## RESULTS

After filtering, 135 occurrence points of *P. caribaeorum* were kept for the analyses (Fig. 2). The model with a lower AIC (2649.8) presented an *Rm* of 3.5 and showed good performance defining occurrence and absence sites for *P. caribaeorum* (AUC $= 0.860 \pm 0.023$) (*Swets, 1988*). The difference between the test AUC and the train AUC yielded a value of 0.015, showing that the model was not over-adjusted and is transferable to novel climate scenarios (*Jueterbock, 2016*; *Warren & Seifert, 2011*).

The environmental variables that contributed the most to the model were salinity (47.9%), concentration of calcite (21%), maximum concentration of chlorophyll-*a* (13.3%), pH (9.2%), and temperature range (8.5%), and were therefore kept in the model (Table 1). The habitat of *P. caribaeorum* is characterized by salinity values higher than 34, a maximum chlorophyll-*a* concentration from 0 to 28 mg/m$^3$, a pH higher than 7.85, and temperature variation up to 10 °C, although these intervals become much narrower when variable correlations are considered. For calcite, the model pointed to a modulating influence, as no minimum or maximum values were indicated.

The output of the HadGEM2-ES climate model for salinity, pH, and temperature (used to calculate temperature range) was used to project the SDM model to novel scenarios for the year 2100, while calcite and chlorophyll-*a* were kept constant. A max SSS equal to 25.97 was used as a threshold to infer suitable areas, thus identifying the potential distribution of the species.

Five hundred forty eight square kilometers were characterized as currently suitable for *P. caribaeorum* (Fig. 3). Under the RCP 2.6 scenario (Fig. 4), the southern range of *P. caribaeorum* distribution would lose suitability, as would the Caribbean, Bermudan, and Floridian regions; however, the total suitable area is close to the one observed for the present: 492,000 km$^2$. An increase in suitable areas in the Guianan, Amazonian, Bahamian, and Caribbean regions was also observed, but the biggest change occurred in the northern part of its distribution, in the northern Gulf of Mexico and along the Carolinian and Virginian regions. In these areas, the potential distribution of the species reached almost as far north as Chesapeake Bay. The southernmost area capable of retaining suitability for the species was near Cape Frio, in southeastern Brazil.

The projection under RCP 4.5 (Fig. 5) also shows that part of the Brazilian coast would lose habitat suitability, only remaining suitable around the Abrolhos Bank and in

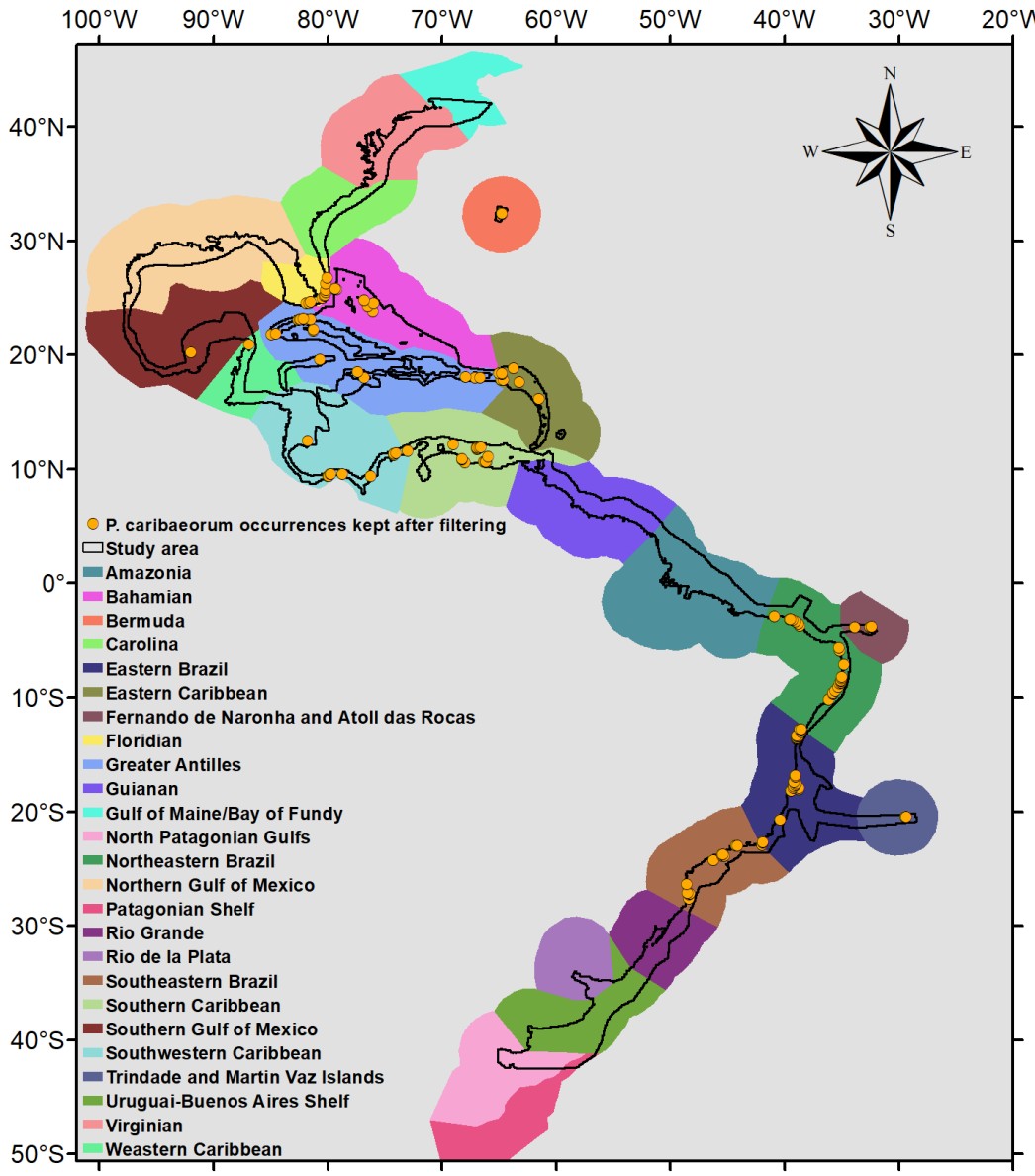

**Figure 2  Occurrence points and Marine Ecoregions included in the study area.** Study area and occurrence points of *P. caribaeorum* after filtering, with the Marine Ecoregions of the World as a reference (*Spalding et al., 2007*).

northeastern Brazil. In the north, the species' potential distribution is similar to the 2.6 scenario, only presenting smaller suitable areas in those regions, 470,000 km² in total.

Under the scenario with the higher anthropogenic impact, RCP 8.5 (Fig. 6), almost the entire potential distribution for *P. caribaeorum* along the Brazilian coast would be lost, including the northeast region. Under RCP 8.5, the anthozoan would also lose suitable habitats in regions around Florida, the northern Gulf of Mexico, and the eastern and western Caribbean.

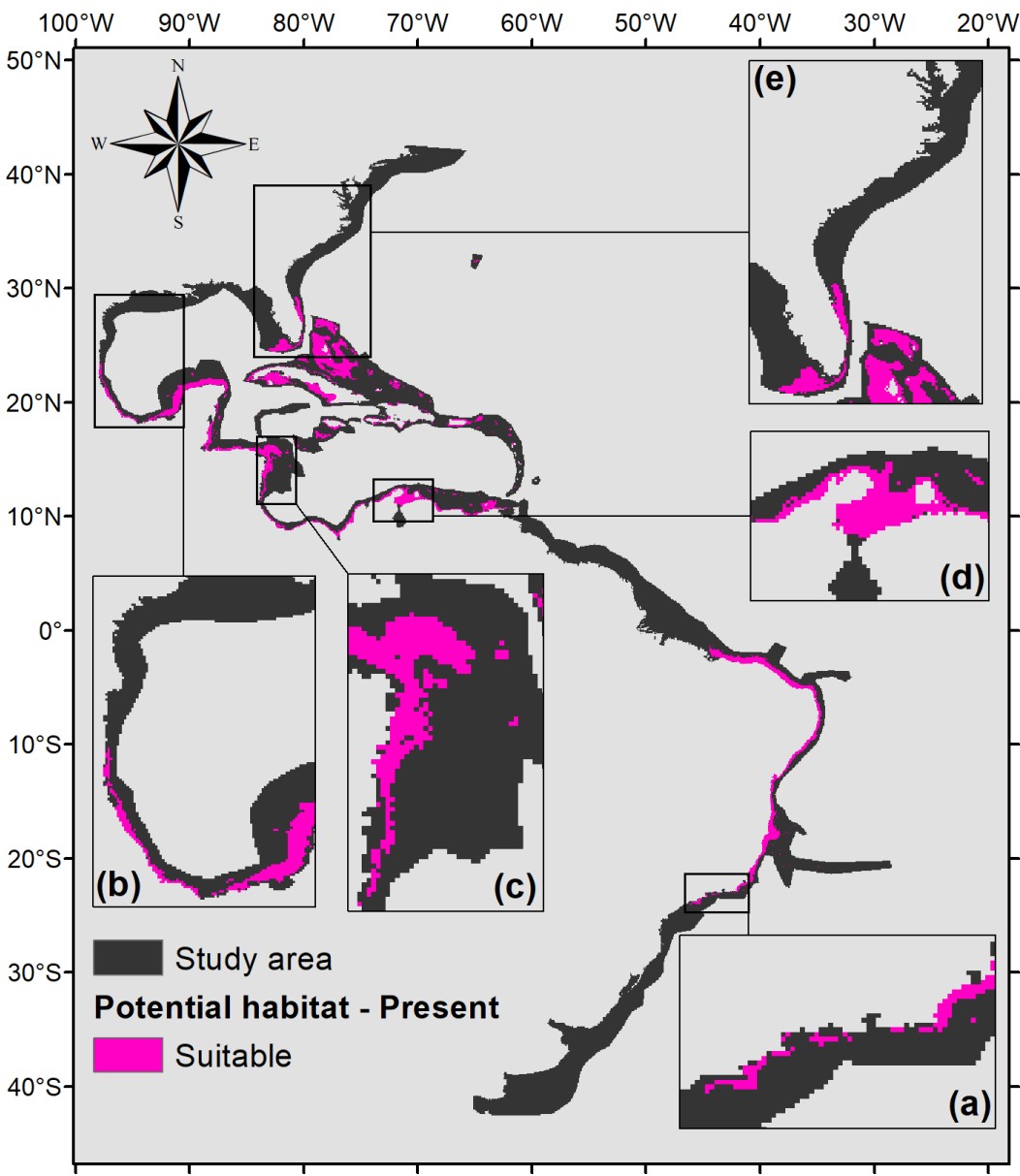

**Figure 3** **Suitable habitat for *P. caribaeorum* in the present.** Details for (A) southeastern Brazil, (B) southern Gulf of Mexico, (C) southwestern Caribbean, (D) southern Caribbean, and (E) Floridian regions.

Overall, for this climate scenario, only one-half of the suitable areas seen in the present, around 275,000 km$^2$, were characterized as a potential habitat for the species, comprising the southern, western, and southwestern Caribbean, the southern Gulf of Mexico, and the Bahamian, Greater Antilles, and eastern Brazil regions. Nevertheless, under the RCP 8.5 scenario, there would be an increase in suitable areas in the southwestern Caribbean, along Nicaragua's coast, similar to the RCP 4.5 forecast (Fig. 5).

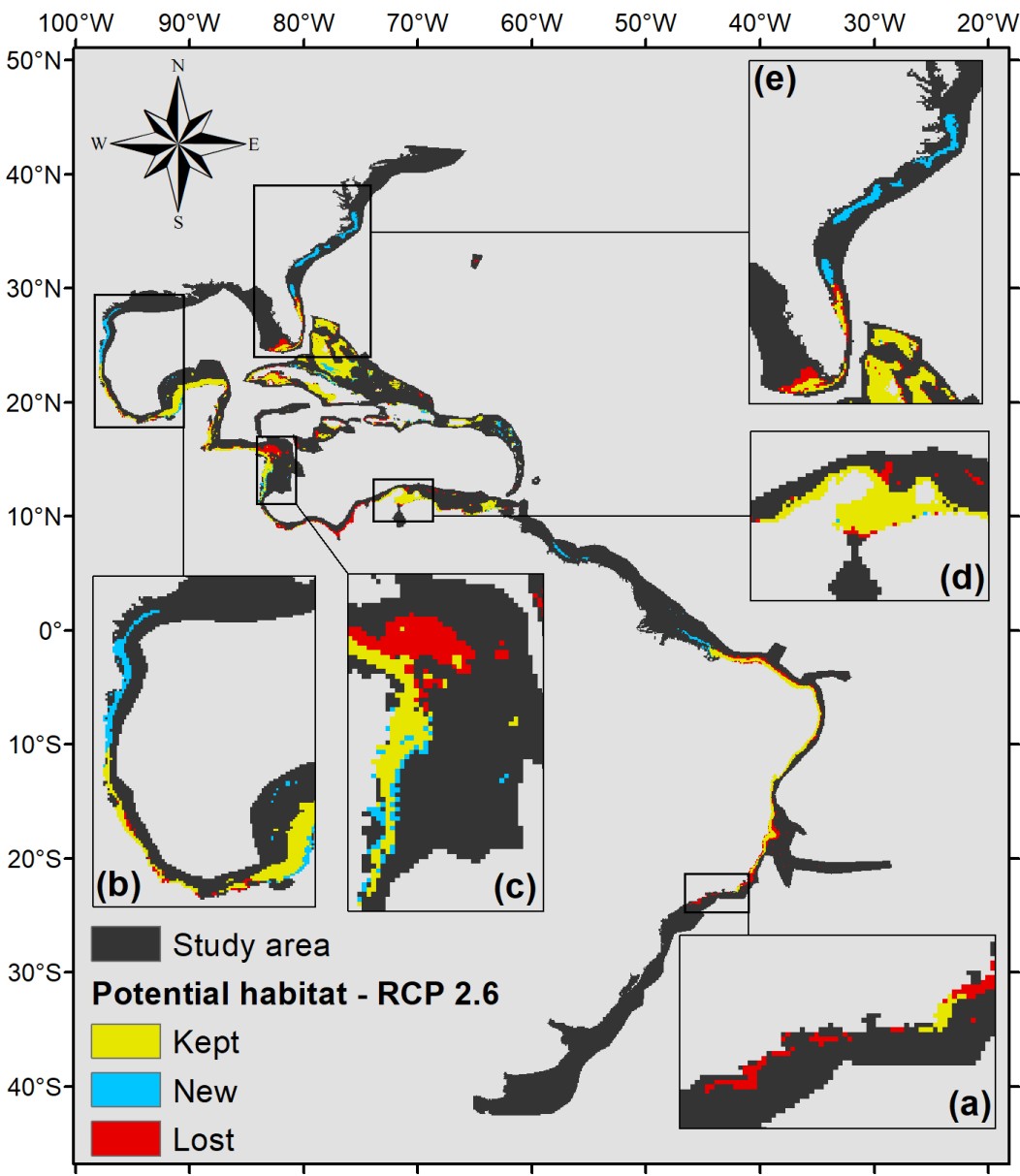

**Figure 4** **Suitable habitat for *P. caribaeorum* for the year 2100 under the RCP 2.6 climate scenario, which includes regions with retained, new, and lost suitability compared with the present.** Details for (A) southeastern Brazil, (B) southern Gulf of Mexico, (C) southwestern Caribbean, (D) southern Caribbean, and (E) Floridian regions.

To demonstrate the most important regions for the species, a map of the intersection of all suitable areas under any climate scenarios for the year 2100 was made (Fig. 7). Figure 7 shows that the northern part of the map (from 0° northward) would have much more significance for the potential distribution of *P. caribaeorum* than the southern part. The suitable areas comprise six ecoregions in the north (southern, southwestern, and western

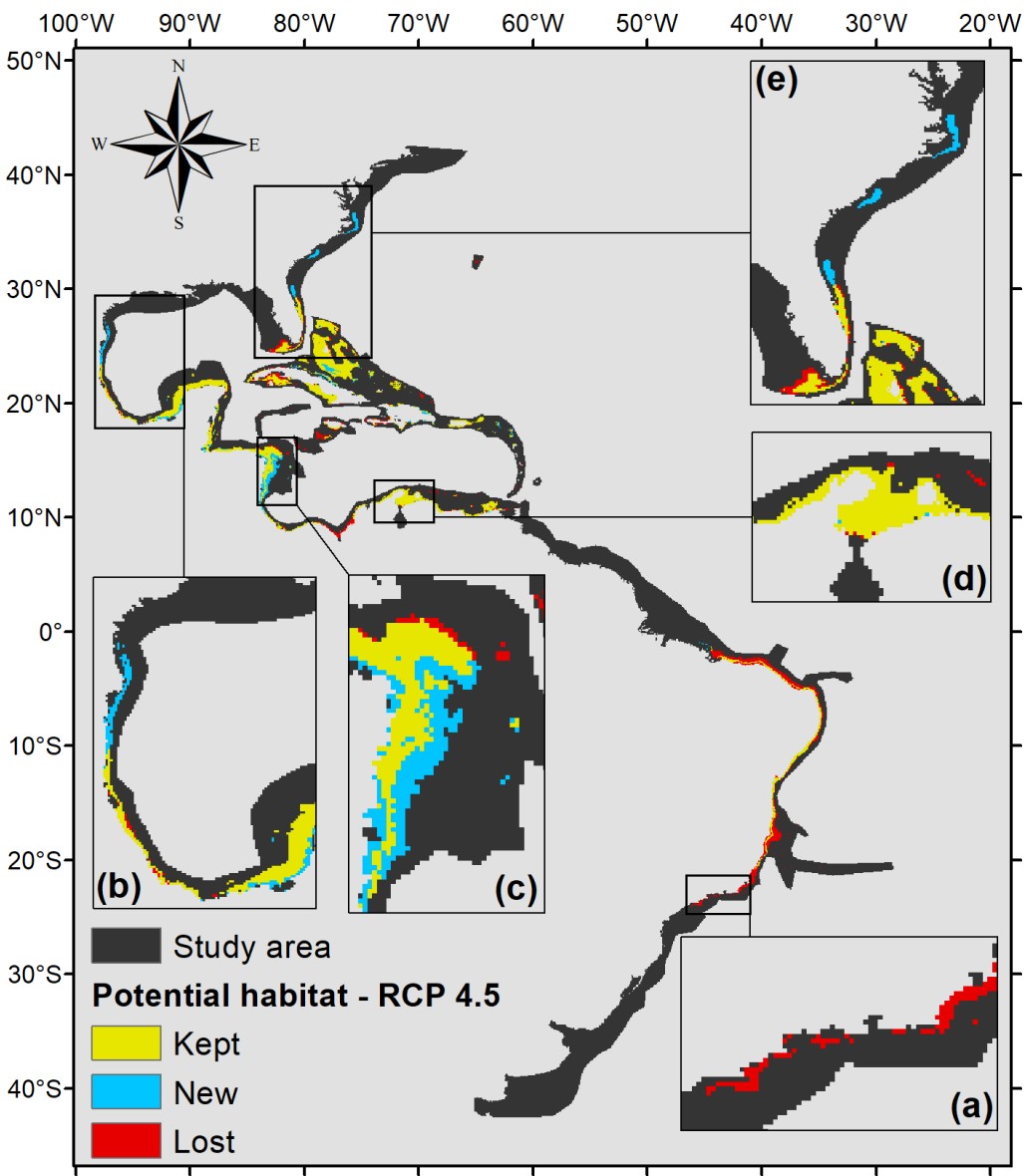

**Figure 5** **Habitat suitability map for *P. caribaeorum*. in the year 2100 under the RCP 4.5 climate scenario.** The map includes regions with retained, new, and lost suitability compared with the present. Details for (A) southeastern Brazil, (B) southern Gulf of Mexico, (C) southwestern Caribbean, (D) southern Caribbean, and (E) Floridian regions.

Caribbean, southern Gulf of Mexico, and Bahamian and Greater Antilles regions) against only two in the south (northeastern and eastern Brazil).

## DISCUSSION

### Environmental variables and *P. caribaeorum*'s habitat

The variables selected to model *P. caribaeorum's* habitat suitability are in line with those used in studies about environmental controls for other anthozoans. For instance, temperature

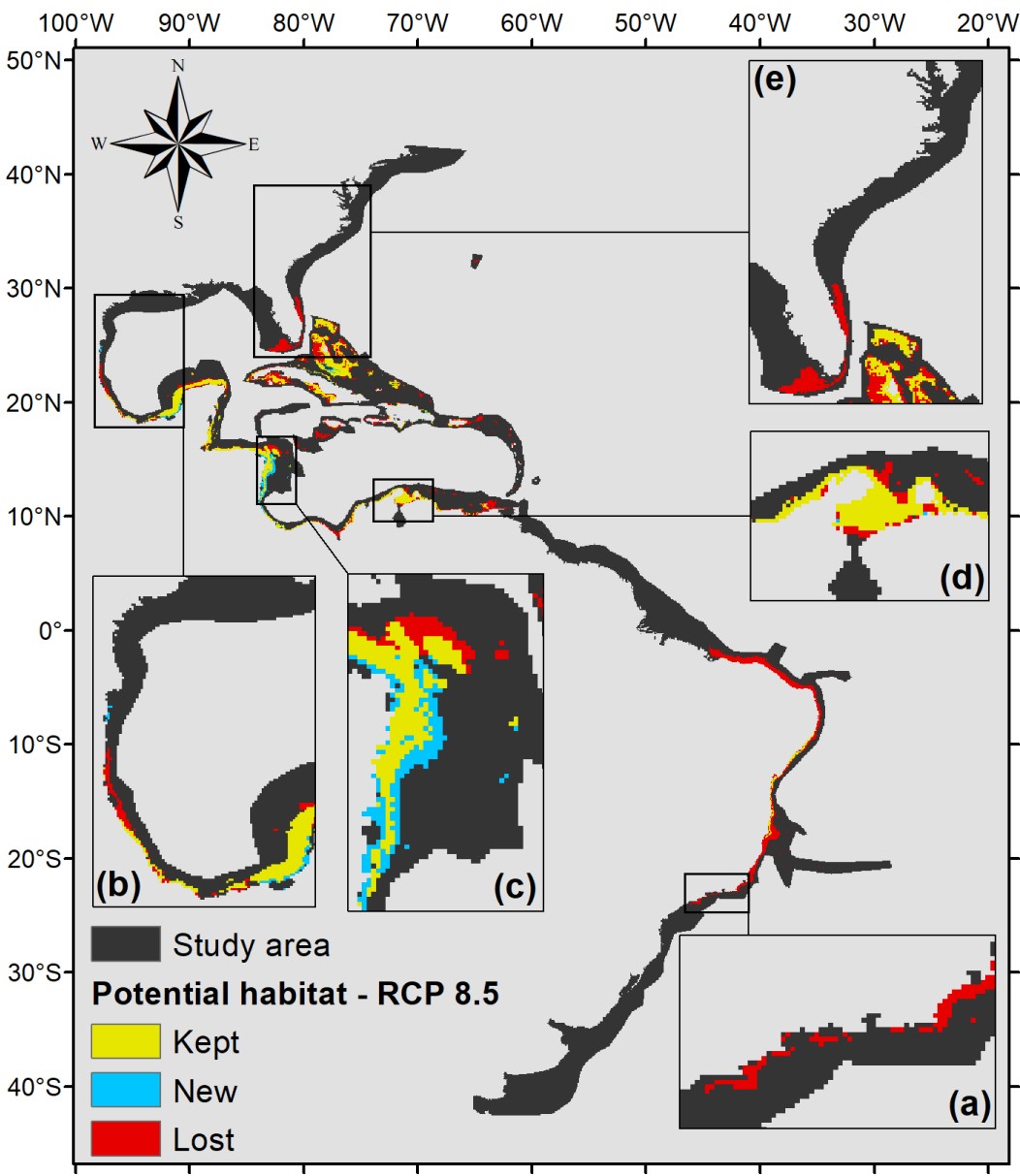

**Figure 6** **Habitat suitability map for *P. caribaeorum*. in the year 2100 under the RCP 8.5 climate scenario.** The map includes regions with retained, new, and lost suitability compared with the present. Details for (A) southeastern Brazil, (B) southern Gulf of Mexico, (C) southwestern Caribbean, (D) southern Caribbean, and (E) Floridian regions.

and salinity are two of the most important factors shaping the growth of modern reefs (*Wood, 1993*). Temperature is also related to coral (*Hoegh-Guldberg, 1999*; *Hughes et al., 2003*; *Villamizar et al., 2008*) and *P. caribaeorum* bleaching (*Kemp et al., 2006*). Variation in salinity may induce osmotic stress and can alter the zonation of zoanthids, including *P. caribaeorum* (*Sebens, 1982*). Changes in chlorophyll-*a* concentrations can also play a role in the distribution of *P. caribaeorum* due to its heterotrophic nutrition and dependency

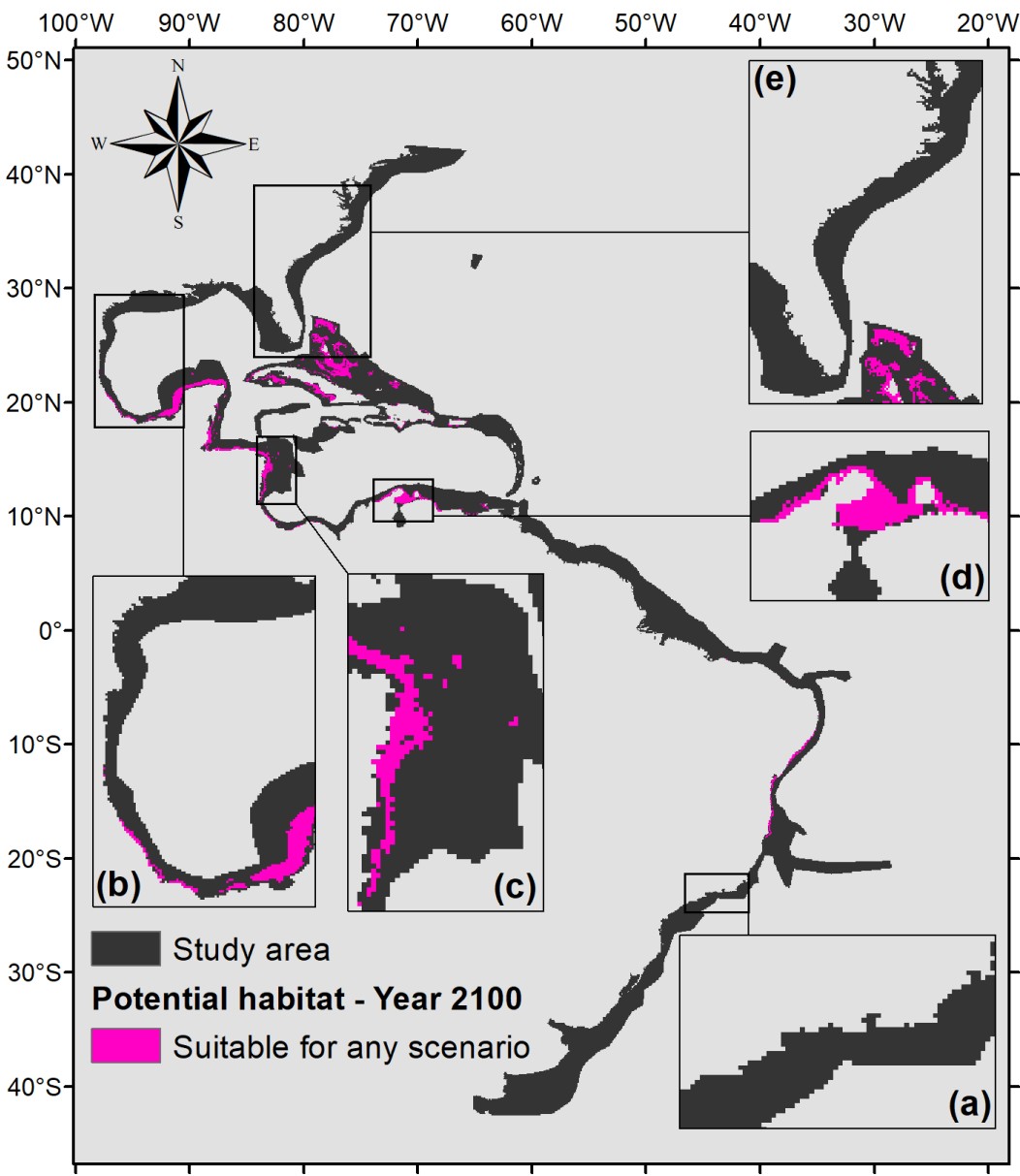

**Figure 7 Map of potential habitat for *P. caribaeorum* in the year 2100 under any climate scenario (pink).** Details for (A) southeastern Brazil, (B) southern Gulf of Mexico, (C) southwestern Caribbean, (D) southern Caribbean, and (E) Floridian regions.

on the water column productivity, unlike most coral reef species, which depend solely on autotrophy and therefore, clear oligotrophic waters. This species also seems to be affected by lower pH (*Kroeker et al., 2013*), which is another consequence of global environmental changes (*Okazaki et al., 2017*).

When the projection of the habitat suitability of *P. caribaeorum* for the present (Fig. 3) is compared with the map of the biggest river discharges in the study area (Fig. 1), it becomes evident that salinity has a major influence on the distribution of this species.

Salinity had a higher contribution to the model, sometimes reducing habitat suitability even when the values of other variables, like temperature and chlorophyll-*a*, were adequate for the species. Freshwater discharges from rivers are also an important source of calcite for oceans; however, calcite is 100 times more soluble in seawater than in freshwater (*Libes, 2009*), so high concentrations of this mineral are not limited to river-influenced regions. In fact, regarding only calcite, habitat suitability is positively correlated with its concentration, possibly because calcite concentration also indicates other important biogeochemical controls.

The maximum concentration of chlorophyll-*a* presented a distribution pattern similar to calcite and was also strongly affected by coastal processes, mainly big river discharges, which fertilize the sea and enhance primary production. From satellite data, chlorophyll-*a* cannot be distinguished from dissolved organic matter (DOM) without an algorithm specific to each region (*Ali et al., 2016*; *Liew, Chia & Kwoh, 2001*; *Vazyulya et al., 2014*). Away from river discharges, chlorophyll-*a* can be re-suspended back into the water column by wind, currents, and waves, and is therefore correlated with suspended sediment (*Walsh, Dieterle & Esaias, 1987*). The values of chlorophyll-*a* data used in this study are relative to its maximum concentration over several years. Hence, it is likely that these values are the sum of all processes listed above: river discharge, new production, and re-suspension. *P. caribaeorum* inhabits coastal regions with high chlorophyll-*a* concentrations, but when these concentrations are related to river discharge, the regions lose habitat suitability due to lower salinity and pH values. This can indicate that despite the low salinity, chlorophyll-*a* and turbidity can be important factors for the species' niche. This information corroborates previous studies (*Segal & Castro, 2011*; *Steiner et al., 2015*), which suggested that low salinity was an exclusion factor for *P. caribaeorum*. However, it is difficult to decouple the influence of variables related to water quality from demands related more to depth and substrate in coastal areas, as both are intrinsically linked in continental shores. For instance, *P. caribaeorum* is quite abundant around remote oceanic islands where eutrophication, turbidity, and other processes typical of continental shores are not present. However, *P. caribaeorum* seems to be more common in regions influenced by coastal processes with high turbidity along the Brazilian coast (I Cruz, pers. obs., 2015).

Similar to calcite and chlorophyll-*a*, pH has a strong coastal component related to salinity, especially around the Amazonian river mouth, where pH drops below 7, and in the Virginian and Carolinian regions, where values exceed 8.25—both regions are devoid of the anthozoan. Regarding pH alone, a habitat with a pH over 7.85 is considered suitable, but due to correlations with the other variables, the habitat of *P. caribaeorum* is constrained to a pH between 8.05 and 8.25.

Lastly, temperature range seems to influence the species in a typical latitudinal fashion, presenting higher values in the northernmost and southernmost regions of the study area due to the intrusion of colder waters onto the continental shelf at higher latitudes. Although *P. caribaeorum* exhibits high tolerance to changes in temperature and water properties in tide pool environments (*Bastidas & Bone, 1996*) and in southeastern Brazil (*Bouzon, Brandini & Rocha, 2012*), extreme variations in temperature and decreases in salinity caused by rain can cause mortality (*Sebens, 1977*). For current environmental

conditions, the latitudes 30° north and 30° south can be recognized as limits to the distribution of this organism, mainly due to the temperature range: between 9 °C and 13 °C in the south and 13 °C and 24 °C in the north. In these regions, salinity reaches a value of 34, with high seasonal variation (*Piola et al., 2000*; *Rasmussen, Gawarkiewicz & Owens, 2005*).

## Present distribution of *P. caribaeorum*

There are regions characterized as potential habitats for *P. caribaeorum* that have no record of the species (Figs. 2 and 3). This indicates either a lack of adequate sampling effort in the Guianan, Greater Antilles, Bahamian, and western and southwestern Caribbean regions, the absence of proper substrata, or other conditions not considered in the present model.

In general, the region between Florida and the Yucatan Peninsula, with the exception of Flower Garden Banks off the coast of Texas (*Tester et al., 2013*), is characterized by the absence of coral reefs due to low temperatures during winter and high rainfall (*Longhurst, 2007*). In shallow areas in the northern Gulf of Mexico previously dominated by corals and sponges, changes in temperature and an increase in river discharge were followed by a community shift around 2005, and these areas are now dominated by algae (*DeBose et al., 2013*). This information corroborates our model predictions for *P. caribaeorum* in that region, strongly influenced by the Mississippi River Delta.

Using *Briggs & Bowen*'s (*2012*) classification, it is evident that all of *P. caribaeorum's* current potential habitat is within warm marine provinces, characterizing the species' distribution as tropical. However, the limits of its distribution are close to warm-temperate and cold provinces, which are characterized as marginal parts of the distribution. Studying reef communities located at higher latitudes can improve our knowledge of how this species will be affected during environmental and climatic changes (*Thomas et al., 2017*); thus, examining *P. caribaeorum's* distribution in these regions is extremely important.

## *Palythoa caribaeorum*'s response to climate change in the year 2100

According to the RCP 2.6 climate scenario for the year 2100, there would be an increase in salinity of 2.5 in the northern Gulf of Mexico, Virginian, and Carolinian regions, in turn increasing habitat suitability for *P. caribaeorum* (Fig. 4). Along the southeastern Brazilian coast, salinity would drop by 0.3, sufficient to become an unsuitable area for the species. Changes would also be seen in the plume of the Amazon river, influencing adjacent coastal regions to a lesser degree.

Values of pH in the year 2100 would drop for all regions, which is one of the factors that could exclude *P. caribaeorum*. Since the anthozoan response to changes in pH is related to salinity, its northernmost distribution should increase due to a weaker influence of rivers and a decline in the temperature range under the RCP 2.6 scenario. On the other hand, decreases in salinity and pH and more intense temperature fluctuations in the southeast and east of Brazil would result in a loss of suitable areas, except along the coast of Cape Frio in southeastern Brazil. This result may be associated with upwelling in this region (*Castro et al., 2006*), which modifies water characteristics and reduces the frequency of osmotic and thermal stress on the species (*Hu & Guillemin, 2016*), making that region more suitable

for the animal. The same process seems to occur along the coast of Colombia, as seen for coral species in that region (*Chollet, Mumby & Cortés, 2010*). The Colombian coast is very important for *P. caribaeorum* since it would remain a suitable habitat under any climate scenario (Fig. 7).

The same trend observed under the RCP 2.6 scenario occurs in the RCP 4.5 scenario, with an increase in potential habitat in the northernmost range of its distribution (Fig. 5). However, for this scenario, *P. caribaeorum* would also lose habitat in the south, including around Cape Frio. Other areas, such as the continental shelf of the Yucatan Peninsula, would gain suitable zones in the RCP 4.5 scenario, due to higher salinity values and lower temperature variations. For regions experiencing a loss of suitable areas, such as the northeastern Brazil and Amazonian regions, the drop in pH values, salinity, and an increase in the temperature range would be responsible.

As expected, the climate scenario with the strongest anthropogenic impact, RCP 8.5, presented the smallest suitable area for the species (Fig. 6). Although it presented smaller temperature fluctuations compared to other scenarios, in the RCP 8.5 scenario *P. caribaeorum* would lose potential habitats in both the northernmost and southernmost ranges of its distribution, without any gain in suitable areas due to a drastic decrease in salinity and pH values. Freshwater intrusions from the La Plata and Amazon rivers would influence the species' habitat along the Brazilian shelf, restricting its distribution to northeastern Brazil. Similarly, the Mississippi River plume would decrease the adequacy for the species in the north. In general, when looking at changes in salinity, pH, and temperature for any climate scenario, the variations are more intense in the north and south via the intrusion of polar and temperate currents into the continental shelf. These differences alter the habitat of *P. caribaeorum* and are more prominent in the Southern Hemisphere.

Regions such as the southern, southwestern, and western Caribbean, the southern Gulf of Mexico, the Bahamas, the Greater Antilles, and northeastern and eastern Brazil would retain suitable areas for the species under any climate scenario (Fig. 7), identifying them as refuges for the species under adverse environmental conditions (*Haffer, 1969*). Except for the Brazilian coast, all of these regions are within the Caribbean Province, which is considered the center of biodiversity in the Atlantic Ocean (*Rocha et al., 2008*; *Briggs & Bowen, 2013*). Under intense climate stress, it is likely that both *P. caribaeorum* and other invertebrate organisms will have their distributions restricted toward this diversity-gathering hotspot in the Atlantic Ocean.

The difference in suitable areas between the Northern and Southern Hemispheres suggests asymmetrical species distribution, with the Northern Hemisphere containing larger suitable habitats. This may result from differences in the oceanographic conditions of both basins. In contrast to other oceans, the Atlantic Ocean presents a net flux of superficial water transport to the north, with the Labrador Current being much weaker than the Malvinas Current; therefore, its principal component is toward the north (*Onken, 1994*). This occurs mainly because of thermohaline circulation and wind, both of which are likely to be altered by climate change (*IPCC, 2014*), in turn helping to remodel the distribution of marine organisms, as predicted in this study.

## Likely changes in reef communities

Studies have shown that *P. caribaeorum* coexists with many other organisms, especially algae such as *Caulerpa racemosa* var. *peltata*, *Bryopsis* spp. (*Magalhães et al., 2015*), *Sargassum*, *Halimeda*, *Laurencia*, *Dictyota*, *Dictyopteris*, *Ulva*, and coralline algae (*Steiner et al., 2015*). Macroalgae are common in coral reefs, sometimes dominating the community and competing for space with corals (*Tanner, 1995*; *Bruno et al., 2009*). The disappearance of *P. caribaeorum* from many Brazilian reefs in the Abrolhos Bank would imply a change in the reef communities. It is possible that with the vanishing of *P. caribaeorum*, colonization of the available space would first be by algae, most likely cyanobacteria, due to their fast growth rate, especially in nutrient-poor environments (*Villaça & Pitombo, 1997*). However, the composition of the new community will depend on multiple factors, from changes in recruitment rates, growth, and mortality to changes in nutrient concentrations and herbivory (*Littler, Littler & Taylor, 1983*; *Sandin & McNamara, 2012*); community resilience is another factor (*Roff & Mumby, 2012*).

Regarding the colonization of new areas by *P. caribaeorum*, the temperate region of the northwestern Atlantic would see its suitable area increased under RCP 2.6 and 4.5, around Cape Hatteras and Chesapeake Bay. Zoanthids have the potential to disperse to great distances due to their long larval stage (*Ryland et al., 2000*; *Polak et al., 2011*) and asexual reproduction (*Acosta, Sammarco & Duarte, 2005*), alongside their potential to raft on floating objects and boats (*Santos et al., 2016*). Thus, it is possible that the colonization of *P. caribaeorum* in the northwestern Atlantic would occur under these climate scenarios for hard bottom environments.

Nicaragua's coast, which is part of the southwestern Caribbean region, was particularly important for the species due to the increase in potential habitat area under RCP 4.5 and 8.5 (Figs. 5 and 6). Few studies were conducted about the benthic community in that region, which presents high coverage of algae of the genera *Dictyota* and *Padina*, corals such as *Orbicella annularis*, *Agaricia* spp., *Porites* spp., *Acropora palmata*, and *Millepora alcicornis*, and lower coverage of sponge and soft corals (*Kramer et al., 2000*). The same study discusses the likelihood of a phase shift in the coming years due to natural disturbances and climate change. Phase shifts from reef-building corals to zoanthids were previously reported in the western Atlantic (*Cruz et al., 2015a*), where they reduced fish diversity associated with the reefs (*Cruz et al., 2015b*), and have the potential to adversely affect regional fisheries and tourism, which are main income sources for the local population.

Although it was assumed in this work, it is not expected that all individuals will be affected in the same way by climate change. It is known that, at smaller scales, differences between symbiont algae (*Kemp et al., 2006*), as well as the anthozoan's plasticity (*Costa et al., 2011*), can result in distinct outcomes for different populations under climate change. Average and maximum temperatures had small contributions to the model and were not considered here; however, it is known that surface waters are likely to experience an increase in temperature, especially in tropical areas (*IPCC, 2014*). The ability to endure such changes will also play a role in *P. caribaeorum's* distribution. The authors highlight here the importance of future experimental work on temperature stress on key organisms, such as *P. caribaeorum,* to better understand the future of tropical reef communities.

It is possible that, due to the scale of the problem, some local variations were not accounted for, such as the discharge of small rivers around the Gulf of Mexico region. Smaller scales and regional studies should use higher resolution data for this matter. Sea level rise can modify wave stress atop coral reefs, favoring the colonization of *P. caribaeorum* over other organisms, including zoanthids (*Rabelo et al., 2015*). This effect was not considered in this study because the IPCC forecast for sea level rise did not surpass 0.8 m, which is smaller than the error of the bathymetric data used in this work.

## CONCLUSIONS

The present habitat of *P. caribaeorum* can be characterized by high salinity values, marine pH, and a small to moderate range of temperature in coastal and shallow environments. The species' current distribution in the western Atlantic Ocean is restricted to warm provinces, identifying it as a coastal and warm-water species, together with shallow water corals. According to this study, this organism's potential distribution is likely to change by the year 2100. It is possible that, under low to mild anthropogenic impacts (RCP 2.6 and 4.5), the species' distribution will be extended northward and will face a decrease of suitable habitat in the southernmost range of its distribution under any climate scenario. Under the scenario with the highest anthropogenic impacts (RCP 8.5), novel suitable areas poleward were not predicted, and only one-half of its potential habitat area would be retained when compared to the present. Under any climate scenario, the Caribbean Sea remained suitable for the species, characterizing this region as a possible refuge. Upwelling regions, such as Cape Frio and northern Colombia, could provide a refuge for this species as well, diminishing the local impacts of climate changes. Although community changes were discussed according to the available literature, biological and oceanographic processes not considered in this study may play an important role in the outcome of these changes.

## ACKNOWLEDGEMENTS

We thank our colleagues Paulo Polito, André Acosta, Romina Barbosa, Bruno Ferrero, and Alexander Jueterbock, each of whom provided advice and expertise that contributed to this work. We would also like to thank Ana Flora Sarti for kindly providing part of the data used in this work, as well as Erica Donlon for the English review. This work is also a product of the SISBIOTA-Mar research network.

### Funding

This research was supported by Fundação de Amparo à Pesquisa do Estado de São Paulo (FAPESP - Process No. 2016/05296-3) and CNPq, through the SISBIOTA network. Igor Cruz was supported by a postdoctoral scholarship (FAPESP - Process No. 2014/17815-0). Tito M.C. Lotufo is also a CNPq research fellow. The funders had no role in study design, data collection and analysis, decision to publish, or preparation of the manuscript.

## Grant Disclosures

The following grant information was disclosed by the authors:

Fundação de Amparo à Pesquisa do Estado de São Paulo: FAPESP - Process No. 2016/05296-3, FAPESP - Process No. 2014/17815-0.

CNPq.

## Competing Interests

The authors declare there are no competing interests.

## Author Contributions

- Leonardo M. Durante performed the experiments, analyzed the data, prepared figures and/or tables, authored or reviewed drafts of the paper, approved the final draft.
- Igor C.S. Cruz analyzed the data, authored or reviewed drafts of the paper, approved the final draft, provided original data and enhanced discussion.
- Tito M.C. Lotufo conceived and designed the experiments, analyzed the data, contributed reagents/materials/analysis tools, authored or reviewed drafts of the paper, approved the final draft.

## Data Availability

All the data used, including matrices, maxent codes, and full list of references used to assemble the matrices, are provided in the Supplemental File.

## Supplemental Information

Supplemental information for this article can be found online at http://dx.doi.org/10.7717/peerj.4777#supplemental-information.

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
