# Peer review of "The effect of climate change on the distribution of a tropical zoanthid (Palythoa caribaeorum) and its ecological implications"

_PeerJ, doi:10.7717/peerj.4777_

## Round 0.1 · original submission · Minor Revisions

Two expert reviewers have evaluated your manuscript and their comments can be seen below. Overall, the comments are positive, but both reviewers have made observations about the manuscript, which need to be taken into account in a revision. Please make sure to answer each comment that has been made and indicate clearly what changes, if any, have been made in the manuscript.

I would like some clarification about chlorophyll a concentrations. For example, in lines 268-270 you mention changes in chlorophyll a concentrations, heterotrophic nutrition and water column productivity. I do not follow the argument, because Coral Reefs are charaterized as oligotrophic and with low chlorophyll a concentrations.

I agree with the reviewer who commented that the discussion is long and should be shortened. I feel that the introduction could also be shortened somewhat.

Also, please check the references as there are a number of issues there. In particular, Rocha et al 2008 seems to have many errors. Authors names are repeated many times throughout and some are even repeated three times right next to each other.

Reviewer 1 ·

Basic reporting

The work is very well written, clear and concise. The bibliography used is updated and the objectives are well thought out and reached.

Experimental design

The experimental design requires some modifications and clarifications:

The authors said that the sampled areas have only one occurrence per pixel (5 arcminutes = 10 km), but you have several records for the same place (< 10 km2) such as Amaral et al., 2009 and Edwards & Lubbock, 1983 for the SPSP Archipelago, the same with Abrolhos bank (Francini filho et al., 2013; Francini filho & Moura, 2010; Segal & Castro, 2011; Villaça & Pitombo, 1997; Bruce et al, 2012). Several records from the same place may overestimate modeling results. You should leave only one record every 10 km2. Check all used records

It must to be clear how many variables (of Twenty-seven layers of environmental data) you really used to assemble the model. List all of them in a Table.

Why did you use the variable concentration of calcite? if the zoanthid P. caribaeorum don't use calcite in your life cycle

Validity of the findings

The article represents an interesting work on a keystone species in the coastal reefs of the Atlantic coast of the Americas. Modeling works on species distributions are important to understand and relate the abiotic parameters that influence the distribution of the same. P. caribaeorum is an excellent organism for this type of studies since it is easy identification and is very abundant in the environments where it inhabits.
As all modeling work is speculative, but the speculations are well based and well discussed, so the scenarios presented have a certain logic within the evaluated parameters. The work was carried out on a global scale and opens the door to work on regional scales in order to better understand the abiotic parameters analyzed in the present study. The results are of great impact for their use in public conservation management policies and establish a baseline for future modeling works with reef benthic communities.

Annotated reviews are not available for download in order to protect the identity of reviewers who chose to remain anonymous.

Reviewer 2 ·

Basic reporting

This is a manuscript on the distribution of the zooanthid Palythoa caribaeorum, both today and as climate change proceeds. I have a few comments:

28 What are coastal processes? This doesn’t follow from the high salinity and small temperature variation.
31 “would lose one-half of its suitable habitats under (what conditions)? Temperature, salinity, pH, etc.
39 One of the problems that is not explained in any article on this species is it’s distribution today: it live on the outer reefs, not the onshore reefs. It seems to me that someone needs to figure out why.
297 The habitat is “composed of coastal regions with high chlorophyll-a concenntrations”. This seems odd. Most coral reefs are in areas that have low nutrients, giving rise to crystal clear waters (= low chlorophyll-a) that allows the sunlight to reach the benthic corals with their zooxanthellae. Can you explain why?
258 The discussion-conclusion goes on for 12 pages. I think the authors can cut this in half?
484 The authors do not say what factors this species responds to. “anthropogenic impacts” means what? What will cause the “decrease of suitable habitat” in the southernmost but not the northernmost range?

Experimental design

This is a manuscript on the distribution of the zooanthid Palythoa caribaeorum, both today and as climate change proceeds. I have a few comments:

28 What are coastal processes? This doesn’t follow from the high salinity and small temperature variation.
31 “would lose one-half of its suitable habitats under (what conditions)? Temperature, salinity, pH, etc.
39 One of the problems that is not explained in any article on this species is it’s distribution today: it live on the outer reefs, not the onshore reefs. It seems to me that someone needs to figure out why.
297 The habitat is “composed of coastal regions with high chlorophyll-a concenntrations”. This seems odd. Most coral reefs are in areas that have low nutrients, giving rise to crystal clear waters (= low chlorophyll-a) that allows the sunlight to reach the benthic corals with their zooxanthellae. Can you explain why?
258 The discussion-conclusion goes on for 12 pages. I think the authors can cut this in half?
484 The authors do not say what factors this species responds to. “anthropogenic impacts” means what? What will cause the “decrease of suitable habitat” in the southernmost but not the northernmost range?

Validity of the findings

This is a manuscript on the distribution of the zooanthid Palythoa caribaeorum, both today and as climate change proceeds. I have a few comments:

28 What are coastal processes? This doesn’t follow from the high salinity and small temperature variation.
31 “would lose one-half of its suitable habitats under (what conditions)? Temperature, salinity, pH, etc.
39 One of the problems that is not explained in any article on this species is it’s distribution today: it live on the outer reefs, not the onshore reefs. It seems to me that someone needs to figure out why.
297 The habitat is “composed of coastal regions with high chlorophyll-a concenntrations”. This seems odd. Most coral reefs are in areas that have low nutrients, giving rise to crystal clear waters (= low chlorophyll-a) that allows the sunlight to reach the benthic corals with their zooxanthellae. Can you explain why?
258 The discussion-conclusion goes on for 12 pages. I think the authors can cut this in half?
484 The authors do not say what factors this species responds to. “anthropogenic impacts” means what? What will cause the “decrease of suitable habitat” in the southernmost but not the northernmost range?

Additional comments

This is a manuscript on the distribution of the zooanthid Palythoa caribaeorum, both today and as climate change proceeds. I have a few comments:

28 What are coastal processes? This doesn’t follow from the high salinity and small temperature variation.
31 “would lose one-half of its suitable habitats under (what conditions)? Temperature, salinity, pH, etc.
39 One of the problems that is not explained in any article on this species is it’s distribution today: it live on the outer reefs, not the onshore reefs. It seems to me that someone needs to figure out why.
297 The habitat is “composed of coastal regions with high chlorophyll-a concenntrations”. This seems odd. Most coral reefs are in areas that have low nutrients, giving rise to crystal clear waters (= low chlorophyll-a) that allows the sunlight to reach the benthic corals with their zooxanthellae. Can you explain why?
258 The discussion-conclusion goes on for 12 pages. I think the authors can cut this in half?
484 The authors do not say what factors this species responds to. “anthropogenic impacts” means what? What will cause the “decrease of suitable habitat” in the southernmost but not the northernmost range?

---

## Round 0.2 · Minor Revisions

I have reviewed your MS and am satisfied with the modifications that have been made. However, I have found some errors that need to be corrected in the attached PDF.

---

## Round 0.3 · accepted · Accept

I am satisfied with the changes made to the manuscript.

#